# Xanthohumol for Human Malignancies: Chemistry, Pharmacokinetics and Molecular Targets

**DOI:** 10.3390/ijms22094478

**Published:** 2021-04-25

**Authors:** Vancha Harish, Effi Haque, Magdalena Śmiech, Hiroaki Taniguchi, Sarah Jamieson, Devesh Tewari, Anupam Bishayee

**Affiliations:** 1Department of Pharmaceutics, School of Pharmaceutical Sciences, Lovely Professional University, Phagwara 144 411, Punjab, India; vanchaharish@gmail.com; 2Department of Experimental Embryology, Institute of Genetics and Animal Biotechnology of the Polish Academy of Sciences, 05-552 Jastrzebiec, Poland; e.haque@ighz.pl (E.H.); m.smiech@ighz.pl (M.Ś.); h.taniguchi@ighz.pl (H.T.); 3Lake Erie College of Osteopathic Medicine, Bradenton, FL 34211, USA; jamieson.sarah93@gmail.com; 4Department of Pharmacognosy, School of Pharmaceutical Sciences, Lovely Professional University, Phagwara 144 411, Punjab, India

**Keywords:** xanthohumol, *Humulus lupulus* L., cancer, proliferation, apoptosis, therapy, prevention, molecular targets

## Abstract

Xanthohumol (XH) is an important prenylated flavonoid that is found within the inflorescence of *Humulus lupulus* L. (Hop plant). XH is an important ingredient in beer and is considered a significant bioactive agent due to its diverse medicinal applications, which include anti-inflammatory, antimicrobial, antioxidant, immunomodulatory, antiviral, antifungal, antigenotoxic, antiangiogenic, and antimalarial effects as well as strong anticancer activity towards various types of cancer cells. XH acts as a wide ranging chemopreventive and anticancer agent, and its isomer, 8-prenylnaringenin, is a phytoestrogen with strong estrogenic activity. The present review focuses on the bioactivity of XH on various types of cancers and its pharmacokinetics. In this paper, we first highlight, in brief, the history and use of hops and then the chemistry and structure–activity relationship of XH. Lastly, we focus on its prominent effects and mechanisms of action on various cancers and its possible use in cancer prevention and treatment. Considering the limited number of available reviews on this subject, our goal is to provide a complete and detailed understanding of the anticancer effects of XH against different cancers.

## 1. Introduction

Tremendous interest has emerged toward bioactive natural compounds owing to their pharmacological activities in various chronic diseases, such as neoplasms, neurological diseases, viral and bacterial infections, and many more. Among different naturally-occurring compounds, hops, which are obtained from the inflorescence (female) of *Humulus lupulus* L. (family Cannabaceae), have gained attention due to their potential bioactive effects; for instance, anti-inflammatory, antimicrobial [1], antioxidant [2], immunomodulatory [3], antiviral, antifungal [4], antigenotoxic [5], antiangiogenic [6], antimalarial [7], diacylglycerol acyl transferase inhibition [8], and anticancer effects [9]. A diverse array of bioactive phenolic compounds, which are prenylated or prenylated flavonoids in nature, are present in hops. The chief prenylated chalcone available in hops is xanthohumol (XH, Figure 1) which is an ingredient used by the beer industry to enhance its aroma and bitter taste [10]. Chemically, XH is written as 3′-[3, 3-dimethyl allyl]-2′,4′,4-trihydroxy-6′-methoxychalcone [11].

The prenylated flavonoids, such as isoxanthohumol, 6-prenylnaringenin, and 8-prenylnaringenin (8-PN), are isomers of XH, which have potent anti-inflammatory, antineoplastic, antidiabetic, estrogenic, antiviral, and antibacterial activities [12]. XH’s structure was identified first in 1957 by Verzele et al. [13], but its advantageous pharmacological properties were not treasured until the 1990s. Prenylflavonoids which are obtained from hops have various biological activities against many ailments, including neoplasms, osteoporosis, postmenopausal hot flashes, digestive issues, neuralgia, toothaches, tension headaches, and earaches [14,15]. In 2007, the Committee on Herbal Medicinal Products of European Medicines Agency reported XH use in conventional medicine for the gentle treatment of symptoms of insomnia and mental stress. Further, the treatment for sleep disturbances, anxiety, and some other diseases by hops has been approved by Commission E of the Germany and European Scientific Cooperative on Phytotherapy [16].

The antiatherogenic activity of XH was investigated by Hirata et al. [17] using an in vivo hamster model. Their findings revealed the anti-atherosclerosis effect of XH is mediated by improved reverse cholesterol transport through efflux of cholesterol from macrophages and excretion to feces. The same study also explored the hepatic transcript analysis of XH, which concluded that XH enhanced the mRNA expression of *cyp7a1* and *abcg8*. The findings of Hirata et al. [17] also concluded that XH decreases apolipoprotein B secretion, prevents oxidation of low-density lipoprotein, and inhibits the synthesis of triglycerides.

Stevens and Page [14] reported the various health applications of XH, including anticancer agents, dietary supplements, antioxidant activities, and estrogenic effects. XH is a wide spectrum chemopreventive phytochemical constituent which shows its activity by inhibiting early-stage tumor growth and metabolic activation of procarcinogens. These procarcinogens, such as 2-amino-3-methylimidazo[4,5-*f*] quinoline, are present in meats, and their activation is prevented by the inhibition of cytochrome P450 enzyme. Stevens and Page [14] confirmed that the dietetic intake of the XH through regular beer is not enough for achieving a chemopreventive effect. However, Magalhaes et al. [4] reviewed the production of XH-enriched beer and its antioxidant, cardioprotective, anti-inflammatory, antiproliferative, cancer chemopreventive, and broad spectrum anti-infective effects. XH mixed with phenethyl isothiocyanate activated Nrf2 and inhibited nuclear factor-κB (NF-κB) in pancreatic cells [18]. Another study by Lin et al. [19] performed various in vitro and in vivo experiments using XH in the context of several chronic diseases. Being a food supplement, XH has a vast array of biological effects and is emerging as a molecule of interest in treating numerous diseases. A recent publication of Iniguez et al. [20] reviewed the effects of XH on nutrition-related noncommunicable diseases. XH has promising effects against cancer by inhibiting angiogenesis, metastasis, and biotransformation of carcinogens as well as influencing programmed cell death and cell cycle arrest. So far, the work published on the health benefits of the XH has provided evidence of various biological effects; however, there is a need to evaluate its toxicity in humans and to develop bioavailability assays for prescribing effective doses. It is also important to study the nutritional content of XH.

Gerhäuser [21] reported the cancer chemoprotective activity of XH with diverse inhibitory mechanisms at various carcinogenesis stages, such as initiation, promotion, and progression. DNA synthesis inhibition, arresting of cell cycle at S phase and activity of enzymes was modulated by XH. Preneoplastic lesions were prevented by XH in organ culture of mouse mammary gland. Jiang et al. [11] reviewed the anticancer activity of XH against various types of cancers and found it was mediated through inhibition of metastasis and carcinogenesis as well as the modulation of key signaling pathways such as suppressing ERK1/2 and inducing reactive oxygen species (ROS). Additionally, it was found that XH can be used as a radio- and chemosensitizer.

Jiang et al. [11] also reviewed the limitations of XH in clinical applications. The major concern is its low bioavailability, which was reported by Nookandeh et al. [22] based on studies conducted on female Sprague Dawley rats, in which the orally administered XH was excreted within 48 h through feces and urine [22]. Pang et al. [23] concluded that the reason behind XH’s poor bioavailability was its localization in the cytosol and ability to bind to cellular proteins. Therefore, attempts must be made to improve the bioavailability of XH by changing the structure and optimizing its physical and chemical properties. It is also reported that interaction of XH with the phosphatidylcholine membrane causes biophysical change in its bilayers.

Ventureli et al. [24] reviewed bioavailability studies of XH. The estimated bioavailability of XH in rats was found to be 33.1% when administered orally and the range of low and high doses were 1.86 and 16.9 mg/kg body weight, respectively. The low solubility of XH is the limiting factor for gastrointestinal absorption. There is limited information available about the bioavailability and bioactivity of XH in humans, but preclinical studies on XH suggests that it has a cancer-preventive effect.

This review will provide complete and up-to-date information on the plausible therapeutic applications of XH against different cancer types. We have also discussed, in detail, the biopharmaceutical limitations of XH that must be resolved, its various important pharmacokinetic aspects, and its novel molecular targets. In this review, we will first highlight, in brief, the history and use of hops as well as the chemistry and structure–activity relationship (SAR) of XH. Then, we analyze the prominent activities of XH on different types of cancers, its mechanism of action, and its possible use to treat and prevent numerous oncologic diseases.

## 2. Description and Utilization of Hops

### 2.1. History of Hops

The joy of drinking beer and art of brewing is over 5500 years old. In 1935, a voyage of archaeologists from the Pennsylvania University Museum and the American Schools of Oriental Science in Mesopotamia discovered a seal baked into pottery that depicted the contents of a brewery vat which was stirred with long poles by two brewery workers. This instance was dated back to 3500–3100 BC [13].

Historical findings concluded that the origin of hops was in Asia, precisely in the lowlands of the Caucasus, southern Syria, and fertile Mesopotamia. From a botanical point of view, China is considered the place of origin as it is the solitary nation globally native to all hop species (*Humulus lupulus* L., *H. scandens* (Lour.) Merr., and *H. yunnanensis* Hu) [25,26,27]. The Slavic tribes were the first to cultivate and use hops for preserving and seasoning beer, beginning in 1500 to 1000 BC. Starting in the 13th century, other countries began to use hops for brewing purposes as well [28]. Pliny (23–79 AD), the great Roman naturalist, referred to hops as “wolf of the willow” because of twinning growth of hops among the willows, proved as “destructive as a wolf to a flock of sheep”. Hops have been cultivated in Germany in the Hallertau region since 736 AD, and in 860 AD, a hop garden was also planted in Nandlstadt [29]. Many monasteries were popular for their hopped beers during the Middle Ages (500 to 1400–1500 CE). The trade of beer has grown drastically due to the expansion of the towns and it was the main drink along with food at that time. In Germany from 1320–30 AD, hopped beer was usually preferred. Later in the 14th century, Belgium’s hop crop grew in importance, and northern German towns benefited from the sale of their hopped beer [30]. Due to the commercialization of hops from the 1990s onwards, the production of hops has been drastically increased.

Promptly, the medicinal possessions of hops were discovered and have made their route into traditional medicine. Conferring to the old herbarium for medicinal treatment, hops were used for the treatment of many conditions, like liver disease, foot odor, leprosy, disturbances in sleep, constipation, and for catharsis of blood. Hops alcohol extracts have been utilized in Chinese medicine for pulmonary tuberculosis and acute bacterial dysentery treatment as well as in Ayurvedic therapies in India [28].

### 2.2. Hop Botany

Hops belong to the *Humulus* genus which is made up of perennial, dioecious, and climbing vines. It is part of the Cannabaceae family which includes the *Cannabis* genus, also known as hemp, marijuana, or hashish [31,32]. There was a belief for years that the representation of genus *Humulus* was mainly by two species, i.e., *Humulus lupulus* L., the common hop, and *H. japonicas* Zieb et Zucc, the Japanese hop [33]. However, *H. yunnanensis* Hu was first described in 1936, and was thought to have originated from the Yunnan province of southern China at high elevations. *H. yunnanensis* was later identified as a third species of its own in 1978 [34]. The popular hop species *H. japonicas* has been an annual plant in Japan, China, and other neighboring islands and is mostly used for gardening purposes [35].

### 2.3. Hop Cultivation

Around 97% of the total cultivated hops worldwide are destined for the brewing industry, and *H. lupulus* L. is the most commonly grown hop for brewing. The production of hops globally is dominated by the USA and Germany, which produce around 70–80% of the world’s supply [36]. Hops are grown throughout most moderate climate regions and its cultivation requires ample sunlight, warm temperatures, high annual rainfall, and fertile soil. The major regions associated with Hop cultivation are given in Table 1.

### 2.4. XH and Brewing

The basic brewing process includes mashing, boiling, fermentation, maturation, filtration, and bottling. The entire brewing process starts with the creation of a mash from malted barley combined with hot water. Malted starch is converted to sugars during this stage. The wort which contains elevated levels of sugar in water is then drawn from the base of mash and poured into a brewing kettle to boil [37,38,39]. During the wort boiling process, bitter taste and palatability to beer, hops are added at different times to build on aroma. The boiled wort is allowed to cool and is aerated. Yeast is then added to the wort for fermentation and generation of byproducts like alcohol and carbon dioxide take place. After fermentation, the green beer undergoes maturation and is then filtered. After filtration, it is carbonated and moved to the holding tank for bottling [4].

Wunderlich et al. [40] developed a technology known as XAN technology which is used during the brewing process to create XH-enriched beer. For unfiltered larger beers, an XH content of around 1–3 mg/L is obtained using this technique, and for filtered dark beers, an XH content of up to 10 mg/L can be achieved. In another study, an increase in XH yield and inhibition of XH isomerization during wort boiling was achieved by using special malts and cereals and the substances generated from the roasting process [30].

Although thought of as a modern invention, hop extracts have been created and used since the 19th century. In 1863, a steam method of hops extraction was patented by Richard Morland. In 1867, another method for extraction of hops using steam was patented by Theophile Breihaupt. Additionally, a method of hop extraction using carbon disulfide alone or a mixture of carbon disulfide and alcohol, ether, or chloroform was patented by John Johnson from Pennsylvania in 1869 [29].

## 3. Chemistry and SAR of XH

The structure of the XH molecule is made up of open-chain flavonoids with trans-configured A and B aromatic rings joined through a three-carbon, α,β-unsaturated carbonyl system substituted by hydroxyl groups at the 4th, 2nd, and 4th positions, methoxy group at the 6th position, and a prenyl unit at the 3rd position (Figure 1).

The biological action of the prenylated chalcones is due to the existence of the α,β-unsaturated keto group. The molecule’s lipophilicity is improved by replacing the A-ring with a prenyl unit and –OCH_3_ group, which gives it a robust affinity for biological membranes [41].

The SAR of XH was studied by Nuti et al. [42] to evaluate the antiangiogenic property by replacing the phenolic group on the B-ring of XH with various substituents provided with different steric and electronic properties like halogens, nitro groups, or methoxy groups. The phenyl group present on ring A was kept unchanged as it was found to be important for antiangiogenic activity. The analogues of the XH were synthesized by substituting various substituents that have shown potent antiangiogenic property, except for the compounds with chlorine and methoxymethyl groups because of poor solubility [42]. The antioxidant activity of desmethylxanthohumol was studied by synthesizing and evaluating its analogues. The SAR study revealed that lower toxicity was shown by the closed ring (cyclization of prenyl group) analogue than desmethylxanthohumol on human umbilical vein endothelial cells (HUVECs) and pheochromocytoma of the rat adrenal medulla (PC12) cells. Better antioxidant property was shown by compounds with 2,3,4-trihydroxyl phenyl and 2,3-dihydroxyl phenyl groups than by their counterparts with 2,4-dihydroxyl phenyl and 4-hydroxyl phenyl groups. Likewise, better antioxidant activity was shown by dimers than by their corresponding monomers. The chalcone with 2,4-dihydroxyl phenyl and 4-hydroxyl phenyl groups, 2,3,4-trihydroxyl phenyl, and the 2,3-dihydroxyl phenyl groups are easily oxidized to ortho-quinone in the H_2_O_2_-induced oxidation procedure and have more potent antioxidant properties than other tested compounds [43].

## 4. Anticancer Potential of XH Based on Preclinical Evidence

Several studies evaluating the potential of XH in cancer treatment have found that it is effective against numerous different cancer models both in vitro and in vivo. Various researchers have examined the anticancer effect of XH in different cancer types which are summarized in Table 2 and Table 3 and discussed in the following sections.

### 4.1. Breast Cancer

The most prevalent form of female cancer worldwide is breast cancer [44,45]. The Notch signaling pathway has significant importance in the normal development of breast cancer cells by deciding cell fate and self-renewal in stem cells [46]. Notch also acts as an oncogene in the growth and progression of breast cancer, and maladaptive amplification of this pathway is associated with increased frequency of breast cancer [47,48]. Thus, inhibition of Notch1 expression can lead to growth suppression and programmed cell death in breast cancer cells [49,50]. The targeting of XH to Notch1 pathway was assessed by two positive controls such as dual antiplatelet therapy (γ-secretase inhibitor) and Notch1 functional assay. The activity of luciferase was determined using a luciferase reporter assay for when Notch1 was bound to the CBF1 transgene. XH reduced the expression of Notch 1, survivin, and Ki-67 and enhanced the expression of caspase-3. The experiment conducted by Sun et al. [51] showed that XH can suppress the growth of breast tumors and promote programmed cell death. Both in vivo and in vitro studies of XH were performed by Sun et al. on breast cancer. The in vivo study used tumorigenicity assays to determine that XH was effective in suppressing tumor growth on a mouse model of breast tumors. The in vitro studies revealed that there is a decrease in notch signaling pathway, and apoptotic regulators such as Bcl-2, caspase-3, and Bcl-extra-large, which was confirmed by MTT assay, Western blot analysis, and flow cytometry [52].

XH orally administered to nude mice which were previously inoculated with breast cancer cells (MCF-7), resulting in central tumor necrosis, a reduced number of inflammatory cells and, an area of focal proliferation, an increased percentage of apoptotic cells, and lower microvessel density. The antiangiogenic effects of XH have also been confirmed and compared with controls via factor VIII expression immunoblotting in XH-treated tumors [53].

Breast cancer is graded based on the expression of progesterone and, estrogen receptors (PR&ER), and human epidermal growth factor receptor 2 (HER2), such as (1) ER+, a positive estrogen receptor; (2) HER2+, a HER2 overexpressor which may be ER+ or ER−; or (3) TNBCs, a triple-negative subtype that does not express any of these receptors. Until now, successful therapies have not been identified for TNBCs [54]. Hs578 T and MDA-MB-231 breast cancer cells are widely used in in vitro studies as cell models for TNBCs [55]. XH was found to inhibit MDA-MB-231 cell proliferation by apoptosis induction through a mitochondria- and caspase- dependent pathway [56]. XH treatment for 24 h repressed the development of MDA-MB-231 and Hs578 T cells with IC_50_s of 6.7 and 4.78 μM, respectively. XH inhibited the invasive phenotype of Hs578 T and MDA-MB-231 cells as well [57]. The uncharacteristic expression of cancerous ALP isoenzymes, which is a biomarker for prognosis of different cancers, has been observed in many malignant tissues [58]. According to the findings of Guerreiro and team [50], the intestinal ALP (IALP) isoenzyme of ALP, which was overexpressed in malignant tissues, was inhibited by XH in MCF-7 cells. However, XH had no significant effect on other types of ALP isoenzyme, i.e., TNS-ALP. Additionally, chemo- and radiosensitizing experiments on MCF-7/ADR cells revealed that XH was able to attenuate MCF-7/ADR cell sensitivity to adriamycin therapies and radiation that inhibited the expression of epidermal growth factor receptor (EGFR), signal transducer and activator of transcription 3 (STAT3), and multidrug resistance mutation 1 (MDR1), also known as ATP binding cassette subfamily B member 1 (ABCB1). These findings indicate that XH could be a powerful chemosensitizer and radiosensitizer, and warrant comprehensive clinical investigation for the potential therapy of breast cancer [59].

### 4.2. Cervical Cancer

Cervical cancer is a prominent initiator of cancer anguish and death in women all over the globe. Nearly two-thirds of examined women are found to have locally advanced cervical cancer, which has poor prognosis [60]. Hence, there is an immediate need for novel, efficient treatments. Yong et al. [61] reported that XH induced apoptosis in the Ca Ski cervical cancer cell line. XH also trigger S phase cell cycle arrest and amplified the activity of caspase-3, caspase-8, and caspase-9. Additionally, the expression of cleaved poly-ADP-ribose polymerase (PARP), apoptosis-inducing factor (AIF), and p53 increased, while concentration-dependent expression of Bcl-2 and X-chromosome linked inhibitor of apoptosis protein (XIAP) was decreased. These findings suggest that apoptosis induced by XH could involve intrinsic and extrinsic apoptotic pathways.

### 4.3. Cholangiocarcinoma

Cholangiocarcinoma is the most common hepatic and biliary neoplasm with 5-year survival rate less than 10%. The anticancer activity of XH was studied on various cell lines of human cholangiocarcinoma (CC-SW-1, SG-231, and CCLP1). XH strongly decreased colony formation, confluency of cells, and cell proliferation. XH showed anti-cholangiocarcinogenic activity, by enhancing the proapoptotic markers, reducing cell cycle regulatory proteins, and suppressing the antiapoptotic markers, enhanced apoptosis and cell cycle arrest. At molecular level, XH reduced the growth of cholangiocarcinoma by inhibiting the Notch1/Akt pathway [62]. There is a crucial role played by STAT3 in the formation of cholangiocarcinoma. XH has the capability to target STAT3 via the Akt-NF-κB signaling pathway, which leads to the inhibition of the proliferation of cells. Therefore, reduction in the activity of STAT3 with 50 µM XH prominently decreased cell growth and enhanced apoptosis when given orally by mixing XH in drinking water and did not exhibit toxicity. Nonetheless, the major limitation is the poor solubility of XH in water [63]. In another study [64], XH exhibited an antiproliferative effect against cholangiocarcinoma by reducing BCL-2 and increasing BAX expression. XH also suppressed BECLIN-1-dependent autophagy, which restricted its toxicity.

### 4.4. Colon Cancer

Presently, colon cancer is one of the leading causes of death in men and women globally. Its incidence is expected to have a significant increase, rising to 1.1 million death by 2030. XH showed antiproliferative effect on various colon cancer cell lines with IC_50_ values of 2.6, 3.6, and 4.1 µM for 24, 48, and 72 h, respectively [65]. A strong cytotoxicity was exerted by XH against HCT-15 colon cancer cell line after a 24 h treatment with an IC_50_ value of 3.6 µM. The studies of Lee et al. [66] recommended XH to be used with other chemopreventive agents to decreased drug resistance by inhibiting the efflux drug transporter MDR1, multidrug resistance protein (MRP) 1 (drug efflux gene ABCC1), MRP2 (drug efflux gene ABCC2), and MRP3 (drug efflux gene ABCC3) to decrease the drug resistance [11]. XH showed anticancer activity in colorectal cell lines SW480 by activating the ataxia telangiectasia mutated pathway and also by enhancing the DNA damage response [67]. XH also demonstrated antitumor activity in the 40–16 colon cancer cell line derived from HCT116 by downregulating Bcl-2 and by activating caspase-3 and caspase-7 [65].

### 4.5. Esophageal Cancer

Esophageal squamous cell carcinoma is the leading cause of deaths globally. The anticancer activity of XH was studied on KYSE30 cell lines in which XH exhibited an antiproliferative effect and foci formation. Keratin 18 (KRT18) was the major target of XH for exhibiting anticancer activity in KYSE30 cell lines. Apoptosis and cell cycle arrest at G1 phase was also major activities of XH and were associated with the markers, such as Bax, PARP (cleaved), cyclin D1 and D3, and cytochrome c [72]. XH also targeted Akt1/2 to suppress esophageal cancer. This targeting mainly involved the downregulation of GSK3β, S6K, and mammalian target of rapamycin (mTOR). XH also decreased the volume and weight of tumors in PDXs which largely express Akt [95].

### 4.6. Glioblastoma

Glioblastoma is a very aggressive brain cancer related with an extremely poor prognosis in adults. There is a great need to develop new treatments due to the developing drug resistance against the presently available chemotherapeutic agents [96]. XH was studied for its potency towards glioblastoma by using T98G cells. It was found that it reduced cell viability and enhanced the apoptosis which involved cleavage of PARP and activation of caspase-3 and caspase-9 in a concentration- and time-dependent manner. Intracellular ROS was also enhanced by XH [97]. XH decreased the viability of U87 cell lines as well. A recent study revealed that XH suppressed the glycolysis through HK2 inhibition, which led to suppression of glioblastoma [75].

### 4.7. Hematological Cancers

Apoptosis and growth arrest in B-acute lymphocytic leukemia (ALL) cells was induced by XH. Moreover, equal cytotoxicity was found when XH was used against adriamycin resistant ALL (L1210) cells. Additionally, prolonged XH exposure to ALL cells improved their sensitivity towards chemotherapeutic medications [76]. An increase in animal lifespan was observed by administration of 50 µg/mouse/day XH in 200 µL PBS in ALL-like xenograft mouse model, and it substantially deferred the insurgence of neurological disorders. XH produced anticarcinogenic activity on various cell lines of cancer like ALL cells and chronic myeloid leukemia (KBM-5) cells by decreasing the activation of NF-κB through the modification of IKK and p65 [77]. There was no drug resistance to XH, though drug adaptation, categorized by the downregulation of Akt, FAK, and NF-κB activities, rendered cells less invasive and more vulnerable to cytotoxic drugs [11,76]. The proliferation, resistance to apoptosis, and transformation of leukemic cells is mainly due to activation of the PI3K/Akt and NF-κB signaling pathways via oncogenic Bcr-Abl tyrosine kinase in Bcr-Abl(+) myeloid leukemia cells, and this Bcr-Abl expression in the Bcr-Abl(+) myeloid leukemia cell line and K562 cells was inhibited strongly by XH [78].

### 4.8. Laryngeal Cancer

Laryngeal squamous cell cancer is one of the most common cancers of head and neck in the US population and has low survival rate [98]. XH has the potential to decrease the viability of laryngeal squamous cells RK33 and RK45 with less side effects [79]. In SCC4 cells, XH exerted strong cytotoxic effect by inhibiting the expression of Bcl-2 and Mcl-1, and also by activating AIF, p53, and PARP [80].

### 4.9. Liver Cancer

XH was studied for its effects and mechanisms of action on hepatocellular carcinoma cellular cell lines (Hep3B, HepG2, and SK-Hep-1) by evaluating cell viability, colony-forming ability, and cell proliferation [99]. At a concentration of 5 µM and higher, XH expressively reduced the cell viability and colony-formation ability in the hepatocellular cell lines. Hepatocellular cancers were treated by inhibiting the Notch signaling pathway which was supported by reduced Notch1 and HES-1 protein expression [82]. The hepatoprotective activity of XH was found to be concentration-dependent. The expression of the proinflammatory factors like MCP-1 and type I collagen a profibrogenic gene was reduced by XH in hepatic stellate cells (HSC) [99].

The cytotoxic and antiproliferative activity of XH and its non-estrogenic derivatives were investigated by Logan et al. [100] on cell lines of hepatocellular carcinoma. Although XH has potent anticancer activity, its use is limited because of its metabolism by gut microbiota and the host’s hepatic cytochrome P450 enzyme, which transforms it to phytoestrogen 8-PN, the most potent phytoestrogen. The same study also compared the antiproliferative activity of XH with its derivatives like dihydroxanthohumol and tetrahydroxanthohumol, which are not metabolized into 8-PN. These derivatives showed more antiproliferative activity than XH in hepatocellular carcinoma (Huh7 and HepG2) cells. XH was also studied on hepatocellular carcinoma cells and it was reported that XH was effective at a concentration of 25 µM against two hepatocellular cancer cell lines, HepG2 and Huh7. XH also suppressed the proliferation, migration, IL-8 expression, and TNF-induced NF-κB activity in both the cell lines even at lower concentrations [101].

XH not only possesses a hepatoprotective effect but also showed potential effects against obesity and hepatic steatosis [99]. The involvement of Nrf2 pathway activation followed p53 induction and was probably due to the chemopreventive activity of XH in hepatocytes [102]. Additionally, XH also exhibited antimutagenic effects against different procarcinogens which are activated through cytochrome P450 enzymes [81,103]. When evaluated against hepatocellular carcinoma cells, XH and its non-estrogenic derivatives dihydroxanthohumol and tetrahydroxanthohumol showed potential anticancer effects by the induction of cell cycle arrest at the G0/G1 phase [100]. Another study showed that XH was able to exhibit anticancer effects by inducing apoptosis and growth inhibition via NF-κB/p53-apoptosis signaling pathway in human liver cancer cells [104].

### 4.10. Lung Cancer

Presently, lung cancer is one of the most common causes global cancer mortality. Out of 1.6 million affected people, 1.4 million people die annually with only 15% survival rate. Multidrug resistance is one of the most important reasons for the poor response of the most lung cancer patients to the standard drug regimens of cancer therapy. The major molecular target of the human lung carcinomas is ERK1/2 kinase cascade. XH was demonstrated as a potential suppressor of p90RSK and ERK1/2 kinases, activator of cellular repressor of E1A-stimulated genes 1 (CREG) protein, and inhibitor of phosphorylation of CREG in A549 lung adenocarcinoma cells. XH was also found to be a strong chemotherapeutic agent against lung carcinoma by arresting the cell cycle at G1 phase, inducing apoptosis, increasing caspase-3 activity, upregulating p53 and p21, and downregulating cyclin D1 [105].

### 4.11. Melanoma

Melanocytes and dendritic cells are present in the dermal–epidermal boundary of the skin and mainly synthesize melanin, a biopolymer pigment. Melanin possesses diverse functions that include chemical and toxic drug absorption, neurodevelopment in embryogenesis, determination of the appearance of an organism, aural processing, and protective coloration [84]. Several extracellular stimuli, such as isobutyl-methylxanthine, are responsible for development of melanogenesis. XH at a dose range 0.5–10 µM inhibited melanogenesis induced by isobutyl-methylxanthine in B16 melanoma cells. Tyrosinase genes (tyrosinase, TRP-1 and TRP-2) also significantly influence the melanogenesis by producing melanin pigment and activity of these genes was reduced significantly by XH. Strong cytotoxic activity of XH was observed in SK-MEL-2 melanoma cells, in which XH suppressed the activity of the DNA topo-I enzyme (topoisomerase) that was involved in adjustment of topological structure of DNA. The study revealed that XH may evolve as a novel inhibitor of topo-I. The strong inhibition of DNA topo-I enzyme led to apoptosis and antiproliferative action. Topo-I is the main target for many cancer treatments and XH is also regarded as broad-spectrum chemopreventive agent because of its topo-I inhibition [11].

A concentration-dependent cytotoxic effect of XH was observed on human melanoma cell lines in subtoxic doses. XH suppressed migratory activity, proliferation, and formation of colonies. XH also decreased the liver metastasis of murine B16 melanoma cells in C57/BL6 mice and has evolved as an important chemotherapeutic agent for hepatic metastasis of melanoma treatment [106]. The anti-melanogenesis effect of XH was also studied on human keratinocytes (HacaT) for melanosome degradation, and in normal human melanocytes and MNT-1 human melanoma cells for its anti-melanogenesis effect. In all the cell lines, XH showed prominent inhibition of melanin synthesis and melanosome export, showing its potential as an inhibitor of pigmentation in humans [107].

### 4.12. Oral Cancer

Approximately 2% of all cancers are oral cancer cases, and 90% of them are diagnosed as oral squamous cell carcinoma (OSCC). This is because OSCC emerged from the oral cavities of epithelial mucosa. The survivin protein is highly expressed in the tissues derived from the patients of OSCC and cell lines. XH was found to exert antitumor activity via inhibition of the overexpression of survivin and activating mitochondrial apoptotic signaling both in vivo and in vitro. XH inhibits Akt-Wee1-CDK1 signaling and causes reduction of survivin phosphorylation on Thr34, survivin ubiquitination, and degradation mediated by facilitated E3 ligase Fbxl7. Hence, XH has been emerged as a strong chemotherapeutic agent for oral cancer [86].

### 4.13. Ovarian Cancer

Ovarian cancer is a deadly gynecologic threat in the United States, and the fifth most common cancer. Because of the challenge in early identification, most instances of ovarian cancer are discovered at stage III or IV and have only a 15–20% survival rate [108]. When A-2780 ovarian cancer cells were treated with XH for 2 and 4 days, it was found that XH instigated exceptional cytotoxicity with IC50 estimations of 0.52 and 5.2 μM, respectively [70]. After XH therapy, significant growth inhibition and downregulation of protein expression and Notch1 transcription were discovered in SKOV3 and OVCAR3 ovarian cancer cells [85].

### 4.14. Pancreatic Cancer

Pancreatic carcinoma with pancreatic ductal adenocarcinoma is the major cause of cancer related deaths in the United States because of their dismal survival rate [109]. Therefore, there is an urgency to develop new treatment approaches with minimal toxicities. XH was found to be effective for the treatment of pancreatic cancer with minimal side effects in many pancreatic cell lines, such as BxPC-3 and PANC-1. XH acted by inhibiting phosphorylation of STAT3 and downregulating the expression of target genes (Bcl-xL, cyclin D1, and survivin). XH also increased the apoptosis of pancreatic cancer cells by inhibition of the Notch1 signaling pathway. The anticancer activity has been demonstrated in many cancers with activated Notch1 and STAT3 signaling [11,110]. XH in combination with phenethyl isothiocyanate reduced the proliferation of PANC-1 cells. XH and phenethyl isothiocyanate, in combination, reduced NF-κB activity and enhanced Nrf2 expression and expression of Nrf2-related genes (SOD, NQO1, and GSTP) in pancreatic cancer cells [18].

### 4.15. Prostate Cancer

Prostate cancer (PC) is the third most prevalent cause of death in men worldwide and is the most frequently diagnosed cancer, with a peak incidence in men over 70. PC treatment choices are determined by a number of variables, including the patient’s living standards and cancer characteristics, such as prostate-specific antigen (PSA) level, tumor stage, and tumor aggressive behavior. The majority of high-risk PC patients are tackled with a combination of radiotherapy and hormone therapy, resulting in a high chance of survival [111]. Androgen steroids influence the development and progression of PC and, accordingly, androgen ablation therapy has been used to treat various degrees of disease. Advanced therapy of PC is mainly based on the interference with androgen deprivation therapy (ADT) and AR signaling. The development of the resistance in PC towards ADT and next-generation ADT is mainly due to molecular AR- modifications. AR-V7 (AR splice variant) is the most common AR-modification that leads to resistance, and mutations in the AR gene are the second most common [112]. Due to this, there is an immediate need for the development of anti-adenocarcinogenic agents that do not produce resistance in PC cells.

Numerous studies have found XH to be effective against PC through several different mechanisms. Tumor necrosis factor-related apoptosis-inducing ligand (TRAIL)-induced apoptosis in PC cells was increased by XH [113]. In PC cells, XH also causes cell death that is not dependent on caspases [114]. Additionally, XH induced cell apoptosis by binding with annexin V-FITC, cleaving PARP-1, activating procaspase-3, procaspase-8 and procaspase-9, depolarizing mitochondria, and releasing cytochrome c from mitochondria. The inhibition of prosurvival Akt, NF-κB, p-mTOR, and NF-κB-regulated antiapoptotic Bcl-2 was linked to XH-induced apoptosis [90].

The anticancer activity of XH, such as induction of apoptosis and inhibition of growth, were tested on hormone-refractory and hormone-sensitive prostate cancer cell lines. The high sensitivity of prostate cancer cells towards XH (20–40 µM) was identified using cell growth or viability assays. The tumor cell destruction was primarily accomplished by apoptosis which was shown by binding of annexin V-FITC to phosphatidylserine of PC3 and PC4 cells, PARP-1 cleavage, procaspase activation, the release of cytochrome c, and depolarization of mitochondria. XH also showed a concentration-dependent suppression of cell viability in BPH-1 and PC3 cells, and decreased activation of NF-κB [89]. The above studies give a basis for the evaluation of XH clinically in treating metastatic hormone-refractory prostate cancer [90].

### 4.16. Thyroid Cancer

The incidence rate of thyroid cancer is high in both women and men, with a lower mortality rate. XH can be a potential molecule of interest because of its promotion of iodine uptake of thyrocytes (FRTL-5), which reduces the need for inadequate surgery and radiotherapy. XH decreased the proliferation and suppressed malignant phenotype (ASCL1) concentration-dependently in thyroid carcinoma. However, the rate of suppression was very low, i.e., >50%, following a treatment with 30 μM XH for 4 days [11]. Another study revealed that XH enhanced the apoptotic activity in TPC-1 human thyroid cancer cells at higher concentrations [115]. XH increased the uptake of radioiodine at nanomolar concentrations in FRTL-5 cells after 3 days of activation [116]. The Raf-1 signal was activated by XH, which led to inhibition of ASCL1, an important factor in medullary thyroid cancer development. Therefore, the activation of Raf-1 produced anticancer activity in MTC [92]. Further investigation is required for evaluating the mechanisms of anticancer activity of XH against thyroid cancer.

## 5. Biotransformation and Pharmacokinetics of XH

The human intestinal epithelial cell line was used to better understand the reasons for XH’s low bioavailability. Caco2, was used to study XH uptake, accumulation, and transport in the human body. After seeding the cells with XH, the studies were carried out for 18–21 days, and V_max_ and K_m_ values were determined for the accumulation of XH in Caco-2 cells [23]. It was found that XH was not absorbed by facilitated transport. Instead, absorption was mediated through the unique binding of cytosolic proteins produced in Caco-2 cells, which contributed to its poor oral bioavailability.

The biotransformation of XH was investigated using rat liver microsomes. The rats were pretreated with hepatic microsomes from isosafrole and β-naphthoflavone nonpolar metabolite. It was found that XH forms three polar metabolites which were identified by ^1^H NMR, mass spectrometry, and liquid chromatography analyses as (1) 5″-isopropyl-5″-hydroxydihydrofurano[2″,3″:3′,4′]-2′,4-dihydroxy-6′-methoxychalcone; (2) 5″-(2‴-hydroxyisopropyl)-dihydrofurano[2′,3″:3′,4′]-2′,4-dihydroxy-6′-methoxychalcone; and (3) a derivative of XH with an additional hydroxyl function at the B ring. A nonpolar metabolite of XH was also identified as dehydrocycloxanthohumol [117].

Microsomes of human liver were used to investigate the biotransformation of XH. It was observed that the primary path of the oxidative metabolism is the hydroxylation of a prenyl methyl group which forms hydroxylated metabolites of XH. The major possibility of XH conversion in stomach is to isoxanthohumol and to a strong phytoestrogen 8-PN. These phytoestrogens are formed from desmethylxanthohumol, a direct metabolic product of XH [118]. The excretion of XH was observed through feces when 1000 mg per kg body weight was administered [119]. In total, 22 metabolites were observed in the feces of rat most of which were flavone derivatives and modified chalcones [22]. Approximately 89% of unchanged XH remained in the intestinal tract and only 11% are its metabolites. In rats, the phase II metabolites identified revealed that they are due to demethylation, sulfation, oxidation, and hydration reactions [120]. Metabolites due to multiple biotransformations of the XH may contribute to the biological activities of XH [121].

The pharmacokinetic profile of e XH was studied by administering a single oral dose of XH to the mice at a dose level of 20, 60, or 180 mg/kg. The samples of blood were collected after different time intervals and plasma levels of XH were estimated by LC–MS/MS. The peak plasma concentration of XH has a biphasic absorption pattern, and XH was observed after 1 h and between 4–5 h after administration, and the half-life was found to be 20 h for all the respective doses [114]. The longer half-lives of XH are due to enterohepatic recirculation and slower absorption rate after administration through the oral route in humans. XH can easily assemble in intestinal cells as most XH molecules are connected to cellular proteins [122]. XH is poorly orally accessible due to the unique attachment of XH to cytosolic proteins in the intestinal epithelial cells [123]. Many examinations have been performed for investigation of the interaction of XH with the phosphatidylcholine model membranes using FT-IR, DSC, and fluorescence spectroscopy. Fast transportation through the cell membrane was shown in such experiments [124].

## 6. Conclusions

This review summarized recent progress on XH, that inhibits carcinogenesis and metastasis in many cancers. Traditionally hops are utilized as sedatives, antispasmodics, bitter stomachics, and antimicrobials. Current laboratory examination has shown that chalcones and flavonones from hops possess several therapeutic effects such as anti-inflammatory, chemopreventive, antioxidant, and antiproliferative properties. XH has emerged as a potential candidate for anticancer therapy. However, present studies are not sufficient to evaluate the anticancer activity of XH and more mechanistic and systematic in vivo studies must be conducted for better understanding of its efficacy for clinical development.

XH showed very promising bioactivities in both in vitro and in vivo examinations on various cancers as mentioned in Table 2 and Table 3. It was found that among all the cancers studied, XH had the most prominent therapeutic effect on the breast cancer, colon, and prostate cancer.

The development of a broad range of human cancer cell lines, such as ovarian, breast, colon, blood, and prostate cancer lines, were inhibited by XH by inducing apoptosis and by inhibiting apoptosis, and modulating various oncogenic signaling pathways (Figure 2). Several vital signaling pathways were tangled in the anticancer activity of XH, such as ROS, ERK1/2, NF-κB, and Akt. Many in vitro examinations were conducted on various cell lines for evaluating the antitumor activity of the XH on different cancers. These studies concluded that antitumor activity of XH occurs through NF-κB, IKK, and p65 inhibition in blood cancer, whereas in breast cancer, XH decreased the expression of Ki-67, survivin, and Notch 1, enhanced the expression of caspase-3, and inhibited EGFR, MDR1, and STAT3. In hematological cancer, there was downregulation of FAK, Akt, and NF-κB activity as well as strong inhibition of Bcr-Ab1 expression. In cervical cancer, apoptosis was induced by reducing PARP and p53 expression. The therapeutic activity of XH on ovarian cancer was due to downregulation of Notch1 and protein expression. Apoptosis in prostate cancer was brought by TRAIL as well as the depolarization of mitochondria by activation of procaspase-3, procaspase-8, and procaspase-9. XH regulated the MCP-1 protein and reduced the expression of Notch1 and HES-1 protein in liver cancer.

Limited in vivo examinations have been conducted to evaluate the anticancer activity of XH on different cancers (Table 3). Various xenograft mouse models were used for studying the antitumor activity of XH. Another major challenge associated with XH is its low oral bioavailability and the conversion of XH to its estrogenic derivative. To date, there are few human studies available for evaluating the therapeutic dose and toxicity of XH. Additionally, there is need to develop bioavailability assays, dose–response curves, and perform molecular modifications to the core chemical structure of XH to have a deeper understanding of its pharmacokinetic and pharmacodynamic profile. In view of the encouraging results presented here, XH seems to be a multitargeted agent and appreciable candidate for drug development for the prevention and treatment of cancer.

## Figures and Tables

**Figure 1 ijms-22-04478-f001:**
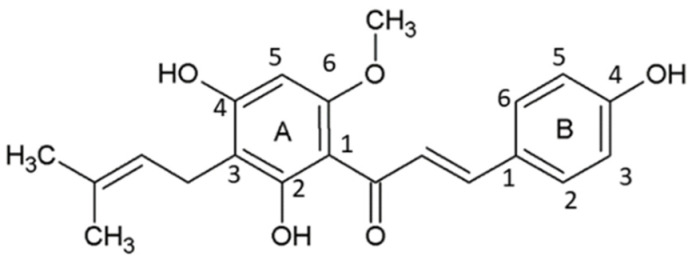
Chemical structure of xanthohumol depicting A and B rings.

**Figure 2 ijms-22-04478-f002:**
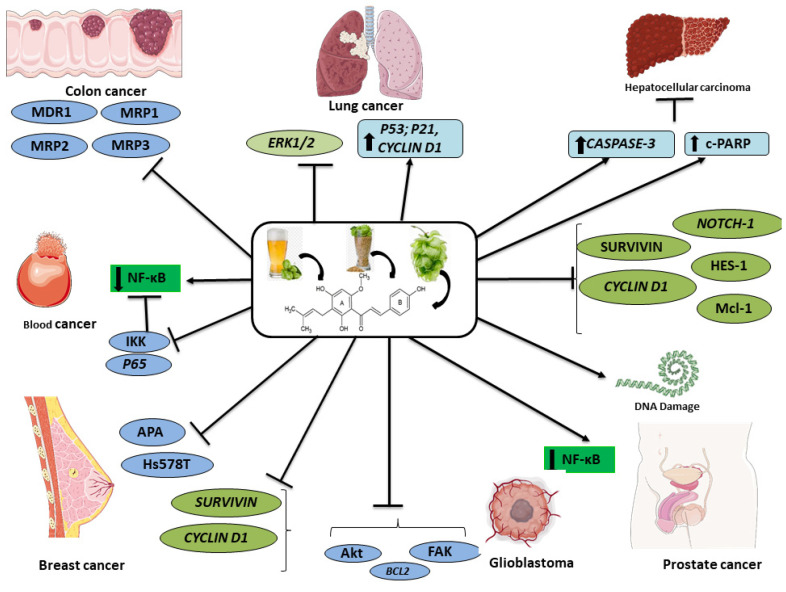
Mechanistic profile of xanthohumol on various cancers. Abbreviations: APA, alternative polyadenylation; Bcl-2, B-cell lymphoma 2; c-PARP, cleaved poly (ADP-ribose) polymerase; ERK1/2, extracellular signal-regulated protein kinase; FAK, focal adhesion kinase; HES-1, split homolog 1; Hs578T, aneuploid mammary epithelium; IKK, IκB kinase; MDR1, multidrug resistance mutation 1; MRP1; multidrug resistance protein 1, MRP2, multidrug resistance protein 2, MRP3, multidrug resistance protein 3; Mcl-1, myeloid leukemia cell differentiation protein 1; NF-κB, nuclear factor-κB.

**Table 1 ijms-22-04478-t001:** Leading hop producing countries in the world.

Country	Region
Germany	Hallertau region
USA	Washington, Oregon, and Idaho
Other countries	Poland, Czech Republic, South Africa, England, Slovenia, Ukraine, China, Australia, and New Zealand

**Table 2 ijms-22-04478-t002:** Antineoplastic effects and underlying mechanisms of action of XH based on in vitro experiments.

Cancer Type	Cell Line	Effects	Mechanisms	Conc.	Reference
Breast cancer	Hs57BT and MDA-MB-231	Decreased cell viability, cell invasion and proliferation	None	4.78–6.7 µM	[57]
	MDA-MB-231	Decreased cell viability	↑Caspase-3; ↑caspase-9; ↓Bax	10 and 20 µM	[56]
	MCF-7	Decreased proliferation	↓ALP isoenzymes	10 µM	[50]
	Adriamycin-resistant MCF-7	Decreased cell viability, stemness, and increased radio- and chemosensitivity	↑Apoptosis; ↑γ-H2AX; ↓STAT3; ↓MDR1; ↓EGFR	10 µM	[59,68]
Cervical cancer	Ca Ski	Decreased proliferation	↑Apoptosis; ↑caspase-3; ↑caspase-8; ↑caspase-9; ↑cell cycle arrest; ↑p53; ↓XIAP	59.96 µM	[61]
Choliangiocarcinoma	KKU-M139 and KKU-M214	Decreased cell growth	↓STAT3	20 and 50 µM	[63]
Colon cancer	40-16 colon cancer	Decreased proliferation	None	4.1, 3.6 and 2.6 µM	[65]
	HT-29 and CDD-18Co	Decreased cell viability	↑Apoptosis; ↑caspase-3; ↑caspase-9; ↓cyclin B1; ↓MEK/ERK; ↓G2/M phase of cell cycle	10 and 100 µM	[69]
	HT-29	Decreased cell viability	None	48 and 72 µM	[70]
	HCT115	Decreased proliferation	↓ABCC 1,2,3; ↓ABCB1	10.2 µM	[66]
Colorectal cancer	FHC, CCD841, CoN, HT29, SW480, LOVO, HCT116 and SW620	Decreased cell proliferation, cell viability, and colony formation	↑Apoptosis; ↓HK2; ↓glycolysis; ↓EGFR-Akt	25 µM	[71]
Esophageal cancer	KYSE30, KYSE70, KYSE410, and KYSE450	Suppressed proliferation, foci formation, and anchorage-independent colony growth	↓Apoptosis; ↑cell cycle arrest (G_1_ phase); ↓Bax; ↓cyclin D_1;_ ↓cyt. c; ↓cleaved-PARP; ↓Bcl-2; ↓cyclin D_3_; ↓KRT18	0.3, 0.6, 1.25, and 2.5 µM	[72]
Glioblastoma	U87 glioblastoma	Decreased cell viability	↑Apoptosis; ↓IGFBP2/Akt/Bcl‑2; ↑mIR-204-3p; ↑ERK/c-Fos	25 µM	[73]
	T98G	Decreased cell viability	↑Apoptosis; ↑ROS; ↑p-p38; ↓p‑ERK1/2; ↑cleavage of PARP ↓caspase-3; ↓caspase-9	20 µM	[74]
	LN229, T98G and U87-MG	Inhibited proliferation, viability, and colony formation	↓Akt-GSK3β-FBW7-c-Myc protein, ↓HK_2_ protein	2, 5, and 10 µM	[75]
Hematological cancers	Acute lymphoblastic leukemia L1210 and adriamycin-resistant L1210	Decreased cell viability, invasion and migration	↑Apoptosis; ↓Akt; ↓FAK; ↓NF-κB	2.5, 5, and 10 µM	[76]
	Chronic myeloid leukemia KBM-5	Suppressed invasion	↑Apoptosis; ↓IKK activity; ↓p65 nuclear translocation; ↓IκBα degradation and phosphorylation; ↓TRAF-2; ↓cIAP-1; ↓cIAP2; ↓survivin; ↓XIAP; ↓Bcl-xL	50 µM	[77]
	Bcr-Abl+ myeloid leukemia cells K562	Decreased adhesion to endothelial cells, cell viability, and invasion	↑Apoptosis; ↓MMP-2; ↓Bcr-Abl; ↑p21; ↑p53	2.5, 5, and 10 µM	[78]
Laryngeal cancer	RK33 and RK45	Decreased cell viability	↑Apoptosis; ↑caspase-3; ↑caspase-8; ↑caspase-9; ↑p53; ↑p21; ↓cyclin D1; ↓ERK1/2	12.3 and 22.5 µM	[79]
	SCC4	Decreased proliferation	↑Apoptosis; ↑PARP; ↑p53; ↑AIF; ↓Bcl-2; ↓Mcl-1	20, 30, and 40 µM	[80]
Liver cancer	HepG2	None	None	10 µM	[81]
	Huh7, Hep3B, SK-Hep1, and HepG2	Decreased colony forming, cell viability and confluency ability	↓HES1; ↓Notch1 pathway	5 µM	[82]
	Hep3B and HA22T/VGH	None	None	108 and 166 µM	[83]
Melanoma	B16	Decreased IBMX-induced melanogenesis	↓Tyrosine enzyme activity	0.5, 1.5, and 10 µM	[84]
	SK-MEL-2	Decreased proliferation	↓DNA topoisomerase 1	14.4 µM	[66]
Ovarian cancer	A-2780	Decreased proliferation	None	0.52 and 5.2 µM	[70]
	OVCAR3 and SKOV3	Decreased proliferation	↓Notch1 pathway; ↑p21; ↑cell cycle arrest	10, 20, and 30 µM	[85]
Oral squamous cell carcinoma	OSCC	Decreased cell viability and reversed radioresistance	↓Survivin; ↑mitochondrial apoptotic signaling; ↓Akt-Wee1-CDK1	1–5 µM	[86]
Pancreatic cancer	PANC1 and BxPC3	Decreased proliferation, viability, and colony formation	↑Apoptosis; ↓p-STAT3	5–100 µM	[87]
	BxPC3, MXPaCa2, and AsPC1	Inhibited cell proliferation	↓NF-κB; ↓VGEF ↓IL-8; ↓mRNA	0.5–25 µmol/L	[88]
Prostate cancer	Hormone-refractory AR^−^PC3	Decreased cell viability	↑Apoptosis; ↓activation of NF-κB	2.5–20 µM	[89]
	Hormone-sensitive AR^+^, hormone-refractory AR^−^ PC3, LNCaP and DU145	Decreased cell viability	↑Apoptosis; ↓NF-κB; ↓p65; ↓p-Akt; ↓p-mTOR; ↓survivin; ↓Bcl-2	24 and 40 µM	[90]
	Hormone-refractory AR^−^ PC3, DU145 PC3, DU145	Decreased proliferation, invasion, and migration	↓p-FAK; ↓p-Akt	2.5, 5, and 10 µM	[91]
Thyroid cancer	MTC (medullary thyroid cancer cells)	Decreased proliferation and malignant phenotype	↑ERK1/2 phosphorylation	10, 20, and 30 µM	[92]

**Table 3 ijms-22-04478-t003:** Antineoplastic effects and underlying mechanisms of action of XH based on in vivo experiments.

Cancer Type	Animal Model	Effects	Mechanisms	Dose (Route)	Duration	Reference
Breast cancer	BALB/c mouse tumor model by using 4T1 cell lines	Suppressed tumor growth; decreased tumor weight and size	↓Survivin; ↑caspase cleavage, ↓Notch-1; ↓Ki-67	100 and 200 mg/kg	14 days	[51]
Colorectal cancer	Xenograft mouse model by using FHC, SW620, LOVO, CCD841, SW480, CoN, HT29, and HCT116	Inhibited tumor cell proliferation	↑Apoptosis; ↑cyt. c release	10 mg/kg (i.p.)	Every two days	[71]
	Male Sprague Dawley rats by using SW480 CRC cells	Inhibited tumor cell proliferation	↑Apoptosis; ↓wnt/β-catenin signaling ↓Bax; ↓ Bcl-2; ↓caspase-3; ↓iNOS; ↓COX-2	5 mg/kg for alternate days	8 weeks	[93]
Esophageal cancer	Patient-derived xenograft mouse model by using KYSE30 cell lines	Decreased tumor volume and weight	↑Apoptosis; Ki-67; ↓KRT18	40, 80, and 160 mg/kg (p.o.)	64 days	[72]
Glioblastoma	Xenograft mouse model by using LN229, U87MG, and T98G cell lines	Reduced tumor weight	↓Akt-GSK3β-FBW7-c-Myc protein	10 mg/kg for every three days (i.p.)	32 days	[75]
Lung cancer	Xenograft mouse model by using HCC827 cells	Suppressed tumor growth	↓Cyclin D1; ↓ERK1/2-fra1 signaling pathway	10mg/kg (i.p.)	32 days	[94]
Pancreatic cancer	Subcutaneous xenograft mouse model by using BXPC-3 cells	Inhibited tumor growth and angiogenesis	↓NF-κB activation ↓tube formation; ↓VGEF; ↓IL8	10 mg/kg/week	5 weeks	[88]

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
