# Peer review of "Xanthohumol for Human Malignancies: Chemistry, Pharmacokinetics and Molecular Targets"

_ijms, 2021, doi:10.3390/ijms22094478_

Round 1

Reviewer 1 Report

The authors review recent progress of "Xanthohumol for Human Malignancies: Chemistry, Pharmacokinetics and Molecular Targets".

A short literature search on xanthohumol and cancer results in more than 2000 references for the last 5 years. The authors cite 120 references.

After the introduction they describe the utilization of hops. A antioxidant and structure-activity relationship description for prenylated chalcones and Xanthohumol follows. After this a review of "Anticancer Potential of XH Based on Preclinical Models" is presented. This part is partitioned for the various malignancies. In  antineoplastic effects and underlying mechanisms of action of XH based on in vitro are presented in a table, and in a separated table, on in vivo experiments.

The review ends with a concideration of "Biotransformation and Pharmacokinetics of XH" and a conclusion. The conclusion "XH seems to be a multitargeted agent and appreciable candidate for drug development for the anticipation and treatment of cancer" is supported by the presented data. 

The review is well structured and easy to read. Thus it seems to be an entrance to the subject.

It would be nice, if all abbrevations would be explained at the first occurrence for example "SAR".

Author Response

The authors of this manuscript express their sincere thanks to the reviewer for the critical assessment of this work. The authors have acted upon the recommendations of the reviewers which have resulted in a significant enhancement in the quality of this manuscript. All modifications incorporated in the manuscript are highlighted in red color font. A “point-by-point” response to each and every comment is outlined below.

Comments:

The authors review recent progress of "Xanthohumol for Human Malignancies: Chemistry, Pharmacokinetics and Molecular Targets".

A short literature search on xanthohumol and cancer results in more than 2000 references for the last 5 years. The authors cite 120 references.

After the introduction they describe the utilization of hops. A antioxidant and structure-activity relationship description for prenylated chalcones and Xanthohumol follows. After this a review of "Anticancer Potential of XH Based on Preclinical Models" is presented. This part is partitioned for the various malignancies. In  antineoplastic effects and underlying mechanisms of action of XH based on in vitro are presented in a table, and in a separated table, on in vivo experiments.

The review ends with a concideration of "Biotransformation and Pharmacokinetics of XH" and a conclusion. The conclusion "XH seems to be a multitargeted agent and appreciable candidate for drug development for the anticipation and treatment of cancer" is supported by the presented data. 

The review is well structured and easy to read. Thus it seems to be an entrance to the subject.

It would be nice, if all abbrevations would be explained at the first occurrence for example "SAR".

Response:

We are thankful to the reviewer for his/her interest and critical evaluation of our manuscript. We are encouraged by the generous comments of the reviewer regarding the quality of our work. As per the valuable suggestions we have explained all abbreviations while mentioning for the first time.

Additionally,

  1. The reference list has been modified and renumbered accordingly. Special attention is given to conform to the order of references and bibliographic style of the journal.
  2. The entire manuscript has been thoroughly checked and edited to ensure uniform style, organization, and quality.

On behalf of my co-authors, I once again express my sincere thanks to the erudite reviewer for the valuable suggestions and constructive input to improve the quality of our manuscript.

Reviewer 2 Report

Dear Authors, your review addresses the most important anticancer activities of xanthohumol in different types of cancer and represents a topical issue in the field of tumor knowledge.  However, I suggest some modifications to improve the relevance of your review.

Major comments:

1- In the introduction, the sentence relating to the observations of Jiang et al. (Line 98-101) regarding the role of XH in tumors is more appropriate to the conclusions paragraph rather than to the introduction. I should delete it or discuss it differently since in the final part of the introduction you say that this review is intended to provide updated information on the role of XH in cancer.

2- Some sentences are written in an unclear form, perhaps due to the lack of verbs. They should be written in a more understandable form. Here are the sentences that in my opinion should be rewritten:

Line 127-131

Line 283-284

Line 399-400

Line 424-425

Line 548-550

3- In Table 2 you should better align the tumor type with the corresponding cell lines and insert separator lines between the different tumor types. Moreover in table 2, Ca Ski instead of Caskl.

The same tip also applies to Table 3.

4- The writings in figure 2 are almost illegible because they are too small. Use a larger font and enlarge the ovals that contain them. Since the figure is set up indicating the molecular targets of XH in the different types of tumors, it is not clear to which type of tumor some of the effects shown in the figure are referred (such as the inhibition of AKT, BCL2 and FAK or of Survivin and Cyclin D indicated in the lower central part of the figure). Moreover, in the central oval I should put the chemical structure or the name of XH rather than the images of beer or hops because the effects shown in the figure specifically concern this molecule.

Along with the abbreviation for MDR1 also enter MRP 1,2, 3

5- In the Conclusions paragraph I would delete the list of tumors (line 543-547) because it is already present in tables 2 and 3 and it is repetitive.

6- You can mention, at least in table, this recent paper: Promotion of ubiquitination-dependent survivin destruction contributes to xanthohumol-mediated tumor suppression and overcomes radioresistance in human oral squamous cell carcinoma.

(Ming Li et al. 2020)Doi: 10.1186/s13046-020-01593-

Minor comments:

  • You should standardize the way of citing other authors. In the text I found author et al. or author and co-workers or author and coworkers or author and team.
  • Line 80-81 performed ….experiments instead of studied….examinations
  • Line 133 Siria instead of Sieria
  • Line 198 delete comma after Although
  • Lines 233-234 The word various is repeated
  • Line 234 delete organ-specific
  • Line 254 the “w” in western is capitalized
  • Line 260 expression instead of functions
  • Line 288 use caspase-3, caspase-8 and caspase-9 instead of CASP-3, CASP-8 and CASP-9 as in the rest of the review.
  • Line 293 insert “the” between is and most
  • Line 296 delete “the” before colony formation
  • Line 313 24h, 48h and 72h instead of 72h, 48h and 24h
  • Line 323 in vivo instead of in vivo
  • Line 377 hepatocellular carcinoma cellular cell lines instead of cell lines of hepatocellular carcinoma
  • Line 434 the s in states is capitalized
  • Line 542 in vitro first than in vivo
  • Line 579 Prevention instead of anticipation

Author Response

The authors of this manuscript express their sincere thanks to the reviewer for the critical assessment of this work. The authors have acted upon the recommendations of the reviewers which have resulted in a significant enhancement in the quality of this manuscript. All modifications incorporated in the manuscript are highlighted in red color font. A “point-by-point” response to each and every comment is outlined below.

General comments:

Dear Authors, your review addresses the most important anticancer activities of xanthohumol in different types of cancer and represents a topical issue in the field of tumor knowledge.  However, I suggest some modifications to improve the relevance of your review.

Response:

We would like to thank the expert reviewer for his/her critical assessment of our manuscript. We have revised our manuscript based on the reviewer’s specific comments as presented below.

Specific comments:

Major comments:

Comment 1:

In the introduction, the sentence relating to the observations of Jiang et al. (Line 98-101) regarding the role of XH in tumors is more appropriate to the conclusions paragraph rather than to the introduction. I should delete it or discuss it differently since in the final part of the introduction you say that this review is intended to provide updated information on the role of XH in cancer.

Response:

We are thankful to the reviewer for this insightful comment and accordingly we have modified the sentence which has been incorporated into the conclusion section (page 14, lines 573-576).

Comment 2:

Some sentences are written in an unclear form, perhaps due to the lack of verbs. They should be written in a more understandable form. Here are the sentences that in my opinion should be rewritten:

Line 127-131

Line 283-284

Line 399-400

Line 424-425

Line 548-550

Response:

We have rephrased the aforementioned sentences for more clarity as follows:

Page 3, lines 128 and 129

Page 6, line 278

Page 11, line 388

Page 11, line 428 and page 12, line 429

Page 14, lines 562 and 564

Comment 3:

In Table 2 you should better align the tumor type with the corresponding cell lines and insert separator lines between the different tumor types. Moreover in table 2, Ca Ski instead of Caskl.

The same tip also applies to Table 3.

Response:

We have aligned the tumor type with the corresponding cell lines and also inserted separator lines between the different tumor types as suggested in both Table 2 and 3 (pages 7-9). In addition Caskl is corrected as Ca Ski (page 7).

Comment 4:

The writings in figure 2 are almost illegible because they are too small. Use a larger font and enlarge the ovals that contain them. Since the figure is set up indicating the molecular targets of XH in the different types of tumors, it is not clear to which type of tumor some of the effects shown in the figure are referred (such as the inhibition of AKT, BCL2 and FAK or of Survivin and Cyclin D indicated in the lower central part of the figure). Moreover, in the central oval I should put the chemical structure or the name of XH rather than the images of beer or hops because the effects shown in the figure specifically concern this molecule.

Along with the abbreviation for MDR1 also enter MRP 1,2, 3

Response:

We appreciate the reviewer’s critical comment. We have modified the figure for style as well as clarity of the presentation. We have enlarged the font size and ovals that contain them. The glioblastoma type of tumor effects shown in the figure are referred with inhibition of Akt, BCL2 and FAK or of Survivin and Cyclin D indicated in the lower central part of the figure is now mentioned in the figure. In addition, in the central oval (now a rectangle), the chemical structure of XH (derived from various sources) is depicted. The abbreviations of MDR and MRP are also inserted in the figure legends (Figure 2, page 15, lines 603-605).     

Comment 5:

In the Conclusions paragraph I would delete the list of tumors (line 543-547) because it is already present in tables 2 and 3 and it is repetitive.

Response:

We have revised the sentence as suggested (page 14, lines 570 and 571).

Comment 6:

You can mention, at least in table, this recent paper: Promotion of ubiquitination-dependent survivin destruction contributes to xanthohumol-mediated tumor suppression and overcomes radioresistance in human oral squamous cell carcinoma.

(Ming Li et al. 2020)Doi: 10.1186/s13046-020-01593-

Response:

We thank the reviewer for suggesting this recent study and according to the suggestion the study has been added as a separate section 4.12. Oral cancer (page 12, lines 444-453) and also in the Table 2 (page 9).

Minor comments:

Comment 1:

You should standardize the way of citing other authors. In the text I found author et al. or author and co-workers or author and coworkers or author and team.

Response:

The citations of other authors are standardized (et al.) throughout the manuscript (page 2, lines 51, 61, 66, 76 and 80; page 3, lines 95, 100 and 109; page 4, line 185; page 5, line 208; page 6, lines 244, 246 and 285; and page 11, line 383). 

Comment 2:

Line 80-81 performed ….experiments instead of studied….examinations

Response:

As suggested, we have written “performed various in vitro and in vivo experiments of XH” instead of studied….examinations (page 2, line 81).

Comment 3:

Line 133 Siria instead of Sieria

Response:

The spelling of Syria is corrected (page 3, line 131).

Comment 4:

Line 198 delete comma after Although

Response:

The comma after although is deleted (page 5, line 198).

Comment 5:

Lines 233-234 The word various is repeated

Response:

The repeated word “various” is rephrased (page 5, line 230).

Comment 6:

Line 234 delete organ-specific

Response:

The word “organ-specific” is deleted (page 5, line 230).

Comment 7:

Line 254 the “w” in western is capitalized

Response:

The “w” in “western” is capitalized (page 6, line 250).

Comment 8:

Line 260 expression instead of functions

Response:

We have used “expression” instead of “functions” (page 6, line 257).

Comment 9:

Line 288 use caspase-3, caspase-8 and caspase-9 instead of CASP-3, CASP-8 and CASP-9 as in the rest of the review.

Response:

We have used caspase-3, caspase-8 and caspase-9 instead of CASP-3, CASP-8 and CASP-9, respectively (page 6, line 287).

Comment 10:

Line 293 insert “the” between is and most

Response:

The word “the” has been inserted between “is” and “most” (page 7, line 293).

Comment 11:

Line 296 delete “the” before colony formation

Response:

The word “the” is deleted before “colony formation” (page 7, line 296).

Comment 12:

Line 313 24h, 48h and 72h instead of 72h, 48h and 24h

Response:

We have revised the sentence as suggested (page 7, line 313).

Comment 13:

Line 323 in vivo instead of in vivo

Response:

We have italicized “in vivo”  as suggested (page 9, line 327).

Comment 14:

Line 377 hepatocellular carcinoma cellular cell lines instead of cell lines of hepatocellular carcinoma

Response:

We have used “hepatocellular carcinoma cellular cell lines” instead of “cell lines of hepatocellular carcinoma” (page 10, lines 373 and 374).

Comment 15:

Line 434 the s in states is capitalized

Response:

We have capitalized “s” in the “United states” (page 12, line 466).

Comment 15:

Line 542 in vitro first than in vivo

Response:

We have written “in vitro” first followed by “in vivo” (page 14, line 578).

Comment 17:

Line 579 Prevention instead of anticipation

Response:

We have used “prevention” instead of “anticipation” (page 16, line 616).

Additionally,

  1. The reference list has been modified and renumbered accordingly. Special attention is given to conform to the order of references and bibliographic style of the journal.
  2. The entire manuscript has been thoroughly checked and edited to ensure uniform style, organization, and quality.

On behalf of my co-authors, I once again express my sincere thanks to the erudite reviewer for the valuable suggestions and constructive input to improve the quality of our manuscript.

Round 2

Reviewer 2 Report

Dear Authors, this version of your manuscript, both from the linguistic as the scientific point of view, is now suitable for pubblication.